# Ventilator-Associated Pneumonia Due to MRSA vs. MSSA: What Should Guide Empiric Therapy?

**DOI:** 10.3390/antibiotics11070851

**Published:** 2022-06-24

**Authors:** Marta Colaneri, Domenico Di Carlo, Alessandro Amatu, Lea Nadia Marvulli, Marta Corbella, Greta Petazzoni, Patrizia Cambieri, Alba Muzzi, Claudio Bandi, Angela Di Matteo, Paolo Sacchi, Francesco Mojoli, Raffaele Bruno

**Affiliations:** 1Division of Infectious Diseases, Fondazione IRCCS Policlinico San Matteo, 27100 Pavia, Italy; marta.colaneri@gmail.com (M.C.); leanadia.marvulli01@universitadipavia.it (L.N.M.); a.dimatteo@smatteo.pv.it (A.D.M.); p.sacchi@smatteo.pv.it (P.S.); 2Department of Biosciences, University of Milan, 20126 Milan, Italy; domenico.dicarlo@unimi.it (D.D.C.); claudio.bandi@unimi.it (C.B.); 3Anesthesiology and Resuscitation Unit, Fondazione IRCCS Policlinico San Matteo, 27100 Pavia, Italy; a.amatu@smatteo.pv.it (A.A.); francesco.mojoli@unipv.it (F.M.); 4Microbiology and Virology Unit, Fondazione IRCCS Policlinico San Matteo, 27100 Pavia, Italy; m.corbella@smatteo.pv.it (M.C.); g.petazzoni@smatteo.pv.it (G.P.); p.cambieri@smatteo.pv.it (P.C.); 5Medical Direction, Fondazione IRCCS Policlinico San Matteo, 27100 Pavia, Italy; a.muzzi@smatteo.pv.it; 6Department of Medical, Surgical, Diagnostic and Paediatric Science, University of Pavia, 27100 Pavia, Italy

**Keywords:** ventilator-associated pneumonia, methicillin-resistant *Staphylococcus aureus*, intensive care unit, negative predictive value

## Abstract

The guidelines on ventilator-associated pneumonia (VAP) recommend an empiric therapy against methicillin-resistant *Staphylococcus aureus* (MRSA) according to its prevalence rate. Considering the MRSA and MSSA VAP prevalence over the last 9 years in our tertiary care hospital, we assessed the clinical value of the MRSA nasal-swab screening in either predicting or ruling out MRSA VAP. We extracted the data of 1461 patients with positive bronchoalveolar lavage (BAL). Regarding the MRSA nasal-swab screening, 170 patients were positive for MRSA or MSSA. Overall, MRSA had a high prevalence in our ICU. Despite the COVID-19 pandemic, there was a significant downward trend in MRSA prevalence, while MSSA remained steady over time. Having VAP due to MRSA did not have any impact on LOS and mortality. Finally, the MRSA nasal-swab testing demonstrated a very high negative predictive value for MRSA VAP. Our results suggested the potential value of a patient-centered approach to improve antibiotic stewardship.

## 1. Introduction

Ventilator-associated pneumonia (VAP) is a frequent infection in the Intensive Care Unit (ICU) setting [1], with a high mortality rate [2].

While a microbiological confirmation is clearly encouraged, on most occasions clinicians have to start an empiric therapy, since delaying treatment is associated with higher mortality [3]. Regarding the initial selection of the antimicrobial regimen, an empiric therapy, based on the awareness of local resistance patterns, should be one of the cornerstones [4,5].

Particularly, while the guidelines recommend an empiric therapy against methicillin-resistant *Staphylococcus aureus* (MRSA) rather than methicillin-susceptible *Staphylococcus aureus* (MSSA) if more than 20% of the Staphylococcus aureus isolates in the local ICU are MRSA [6], there is a paucity of real-life data supporting it. Thus, we may require an individual-based approach for VAP rather than a primarily prevalence-based approach [7]. To go along with this alternative mindset, it is mandatory to identify more targeted risk factors for MRSA infection, such as MRSA colonization [8].

With this in mind, the primary aim of this study was to evaluate MRSA and MSSA VAP prevalence over the last nine years in our hospital, Fondazione IRCCS Policlinico San Matteo of Pavia (IRCCS-FPSMP). The secondary aim was to assess the clinical value of the MRSA nasal-swab screening, in either predicting or conversely ruling out MRSA VAP.

## 2. Results

Overall, data about 1461 positive BAL samples were extracted. Among these, 170 (11.6%) were positive for MRSA or MSSA.

Characteristics of patients with MSSA and MRSA VAP are shown in Table 1. Significantly, there were no differences in ICU between patients with MSSA and MRSA VAP.

Among VAP due to *Staphylococcus* spp., the prevalence of MSSA significantly increased over the years from 56.5% in 2012 to 85% in 2021; by contrast, the prevalence of MRSA significantly decreased from 43.5% in 2012 to 15% in 2021 (*p* = 0.038; Figure 1). However, MRSA had a very high prevalence overall (24.1%). During the considered period, the prevalence of MRSA was higher than 10–20%, that is, the guidelines threshold to define a low-risk ICU for VAP due to this bacterium, or slightly lower, as it happened in 2019 (18.2%) and in the current year (15%). The exceptions are in 2013 and 2017, when prevalence was significantly lower than in the other years (10% and 6.7%, respectively).

Moreover, there was a general downward trend in MRSA prevalence (from 9.4% in 2012 to 1.3% in 2021, *p* = 0.001; Figure 2a), while MSSA remained steady over time (from 12.3% in 2021 to 7.1% in 2021, *p* = 0.218; Figure 2b). Particularly, these trends were unaffected by the COVID pandemic.

With regard to the secondary outcome, among the 1461 patients with VAP, 976 (66.8%) were tested for MRSA nasal colonization before BAL. Accordingly with the fact that the nasal screening for MRSA was not routinely performed before 2016, the majority of the nasal tests, namely, 896, were collected from 2016 onwards. Among the specimens collected from 2016 onwards, only 22 resulted positive for MRSA (2.46%) (Table 2).

The MRSA nasal-swab testing demonstrated a 42.1% sensitivity and 98.4% specificity, with a PPV of 36.4% and a NPV of 98.7% (Table 3).

## 3. Discussion

Our results showed that, despite a downward trend in prevalence of VAP due to MRSA over the last nine years, it has overall remained above 20% in the ICU of IRCCS-FPSMP.

These findings are in line with the ECDC reports, which include countries with both low and high percentages of MRSA and outline a decreasing trend of MRSA infections during the period 2015–2018 [9]. In our case, this trend was notably unaffected by the COVID pandemic. This result is not entirely concordant with the current state of the literature, since the hard COVID-19 knock on our health systems widely increased hospital-acquired infection rates, including MRSA-related infections [10]. However, there are also some well-known and positive consequences of the measures of control and prevention implemented during the pandemic, especially on a microorganism which resides in the nose and on the skin, always covered by masks and gowns [11].

Regarding our choice of using data from positive BAL samples, it has been made according to the clinical practice guidelines on the management of VAP in adults, which recommend both invasive and noninvasive sampling with semiquantitative cultures as the preferred diagnostic methodology [12]. We believe that, although some cases of VAP might have been missed, obtaining microbiological data from BAL samples was crucial for the purposes of our study.

In our study population, MRSA is responsible for only a few cases of VAP, frequently resulting in many unnecessary empiric treatments. MSSA remains more involved in VAP, which warrants greater consideration [13].

In light of the very low absolute prevalence of MRSA as a cause of VAP in our reality and the disproportionate number of patients who would need to be treated unnecessarily empirically, we would really rethink the appropriateness of using the threshold proposed by the guidelines (10–20% relative prevalence of MRSA) [14].

Instead, it might be more meaningful to consider each patient’s individual risk factors for MRSA infection such as nasal colonization, previous antibiotic treatment and prolonged hospital stay, as prompted by recent findings [7]. In our opinion, this individual-based approach might be valuable also in a context where novel multiplex real-time PCR assays on respiratory materials will have a growing role in early diagnosis of VAP [15].

We want to highlight that MRSA nasal colonization, which is a recognized risk factor for MRSA VAP, has a significantly high NPV in our analysis. This finding has been already widely demonstrated in the current literature, also for extrapulmonary MRSA infections [16]. However, it should be mentioned that our result may be driven by the very low prevalence of MRSA VAP in our cohort of patients. This consideration prevents us from surely asserting that MRSA nares screening is a valuable tool to rule out MRSA VAP; though, it validates our initial hypothesis that moving forward the prevalence-based approach is crucial.

The limits of this study are the monocentric and retrospective nature, the lack of data on antibiotic treatments performed, and the clinical comorbidities, which are unfortunately not fully available from the ICD-9CM codes.

## 4. Materials and Methods

### 4.1. Study Design and Participants

This is a single-center, retrospective, observational cohort study.

The SkyNet database is a Relational Database Management System which includes microbiological and clinical data from all the patients admitted to IRCSS-FPSMP.

Data of positive bronchoalveolar lavage (BAL) samples of patients admitted to the ICU from 1 January 2012 to 13 December 2021 were retrospectively extracted from the SkyNet database.

According to the new National Healthcare Safety Network (NHSN) definition [17], only the BAL from patients with signs, symptoms, and imaging test results consistent with pneumonia were included. Moreover, BAL samples without leucocytes and/or inflammatory cells were excluded.

The samples of the MRSA nasal-swab testing of ICU patients with VAP from 1 January 2012 to 13 December 2021 were retrospectively extracted from the SkyNet database. Any patient with at least one positive nasal swab for MRSA at any time before VAP onset was considered colonized by MRSA. MRSA nasal-swab screening was extensively performed on all new ICU admissions from 2016, so only data from that date onwards were considered in our statistical analysis. Since the same year, a nasal decolonization with mupirocin of patients found colonized with MRSA has been routinely performed in our ICU.

### 4.2. Outcomes

The primary outcome was to evaluate MRSA and MSSA VAP prevalence over the last nine years in our tertiary care hospital. The secondary outcome was to evaluate the performance of the MRSA nasal swab in predicting or ruling out MRSA VAP.

### 4.3. Statistical Methods

The trend of positive BAL for MRSA or MSSA over the years and the differences in prevalence were evaluated by Chi-square tests for trend. The sensitivity, specificity, positive predictive value (PPV), and negative predictive value (NPV) of the MRSA nasal swab were calculated.

## 5. Conclusions

In conclusion, our results showed that the trend of MRSA VAP in our ICU has declined over the years, despite the COVID pandemic. Furthermore, we suggested the potential value of a patient-centered approach in order to improve antibiotic stewardship.

## Figures and Tables

**Figure 1 antibiotics-11-00851-f001:**
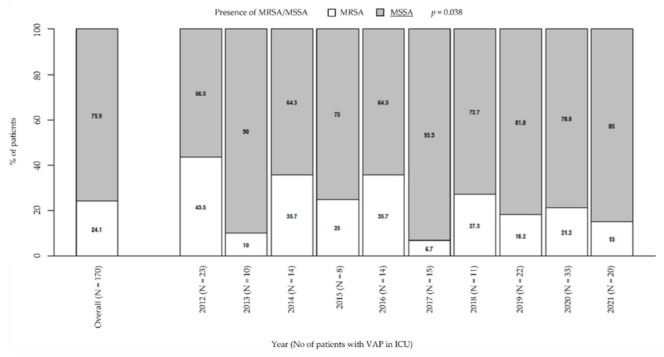
Prevalence of MRSA/MSSA in patients with VAP in ICU. *p*-value by Chi-squared test for trend in proportions. ICU: intensive care unit. MRSA/MSSA: methicillin-resistant *Staphylococcus aureus*/methicillin-susceptible *Staphylococcus aureus*. VAP: ventilator-associated pneumonia.

**Figure 2 antibiotics-11-00851-f002:**
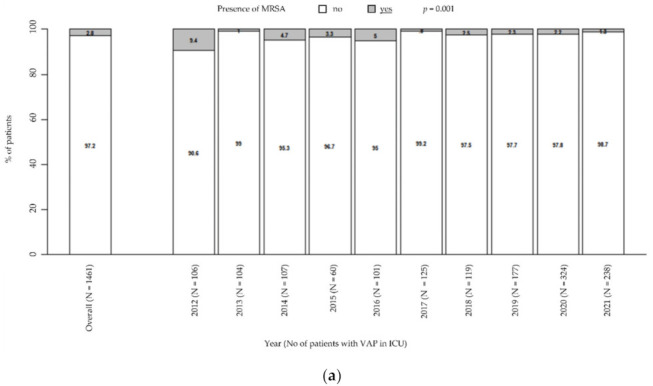
Prevalence of MRSA in patients with VAP in ICU (**a**). Prevalence of MSSA in patients with VAP in ICU (**b**). *p*-value by Chi-squared test for trend in proportions. ICU: intensive care unit. MRSA: methicillin-resistant *Staphylococcus aureus*. MSSA: methicillin-susceptible *Staphylococcus aureus*. VAP: ventilator-associated pneumonia.

**Table 1 antibiotics-11-00851-t001:** Characteristics of patients with MSSA and MRSA VAP.

Variables	Overall (N = 170)	MRSA (N = 41)-Statistics	MSSA (N = 129)-Statistics	*p*-Value
Age (y), median (IQR)	61 (50–72.75)	66 (57–76)	60 (50–70)	0.107
Gender (male), n (%)	114 (67.1%)	26 (63.4%)	88 (68.2%)	0.705
ICU-LOS before VAP onset (days), median (IQR)	3 (0–8)	3 (0–11)	3 (0–6)	0.314
ICU LOS after VAP onset (days), median (IQR)	11 (4–26)	13 (5–23)	11 (4–27)	0.806
Polymicrobial infection, n (%)	81 (47.6%)	16 (39%)	65 (50.4%)	0.276
No. polymicrobial infection, median (IQR)	1 (1–2)	1 (1–2)	2 (1–2)	0.267
Nasal swab (screening MRSA), n (%)	95 (55.9%)	23 (56.1%)	72 (55.8%)	1000
MRSA in nasal swab, n (%), [N = 95]	8 (8.4%)	8 (34.8%)	0 (0%)	0.000
MRSA in blood culture, n (%)	9 (5.3%)	7 (17.1%)	2 (1.6%)	0.001
MSSA in blood culture, n (%)	14 (8.2%)	0 (0%)	14 (10.9%)	0.023

*p*-values by Chi-squared test or Fisher’s Exact test as appropriate for qualitative variables, by *t*-test or Mann–Whitney test as appropriate for quantitative variables. ICU: intensive care unit. IQR: interquartile range. LOS: length of stay. MRSA: methicillin-resistant *Staphylococcus aureus*. MSSA: methicillin-susceptible *Staphylococcus aureus*. VAP: ventilator-associated pneumonia.

**Table 2 antibiotics-11-00851-t002:** MRSA nasal-swab screening.

MRSA Nasal-Swab Screening	Non-MRSA VAP	MRSA VAP	Total
Negative	863	11	874
Positive	14	8	22
Total	877	19	896

MRSA: methicillin-resistant Staphylococcus aureus. VAP: ventilator-associated pneumonia.

**Table 3 antibiotics-11-00851-t003:** Statistical analysis.

Sensitivity, %	42.1
Specificity, %	98.4
PPV, %	36.4
NPV, %	98.7

NPV: negative predictive value. PPV: positive predictive value.

## Data Availability

The data presented in this study are available on request from the corresponding author.

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
