# Peer review of "Ventilator-Associated Pneumonia Due to MRSA vs. MSSA: What Should Guide Empiric Therapy?"

_antibiotics, 2022, doi:10.3390/antibiotics11070851_

Round 1

Reviewer 1 Report

Dear Editor,
Thank you for giving me the chance to review the manuscript titled “Ventilator-Associated Pneumonia due to MRSA vs MSSA: What Should Guide Empiric Therapy?”, submitted for publication in the Journal “Antibiotics”. In this paper the Authors are providing information on MRSA prevalence among their VAP cases, during the last decade. It is a single center study, capturing data on some 1461 VAP cases diagnosed on the basis of a positive BAL culture. They then move on to performing an unadjusted comparison in outcomes between MRSA and non-MRSA patients and they end up by examining the performance of nasal MRSA colonization in predicting the MRSA etiology. They conclude that there is a downward trend in MRSA percentage through the years, which remained rather unaffected by the COVID-19 pandemic, that there is no unadjusted difference in outcomes and that nasal swabbing in a great tool in ruling out MRSA pneumonia. They also praise a personalized approach in VAP empiric treatment, as opposed to following the fixed guidelines suggestions.
While I agree with all the concluding statements, I cannot but be concerned about the methods used to reach these conclusions.
First, the authors used the BAL records as a means to detect VAP cases. Bronchoscopy and BAL might have missed a significant amount of cases, leading to ascertainment bias. I think that the Authors need, at a minimum, to provide some information on how is VAP diagnosed in their institution and whether they expect that BAL detected cases are a representative sample of their total VAP cases.
Second, while the primary objective is probably served by an unadjusted comparison of yearly prevalences (though some assertion that patient populations did not change during this period would be appreciated) this is probably not the case for their secondary objectives, that is the impact of MRSA infection on patient outcomes. The Authors noticed that there were no differences in outcomes between MRSA and non-MRSA infections, however from the few data they provide it is evident that MRSA cases were probably older, while other works point to MRSA patients having also greater disease burden. According to the Methods section, the database that they used has clinical data and, as such, they could use them to build some regression models, thus exploring the true relation between MRSA and outcomes. If this cannot be done, then it should be clearly stated that these are unadjusted comparisons and the significance of the (absence of) finding should be downgraded.
Third, the Authors conclude that absence of nasal MRSA colonization has an extremely high negative predictive value, and thus clinicians can use it to rule out MRSA etiology. I admit that this conclusion is likely enough to be true, I am afraid however that the data presented herein do little to support it. According to Table 2, nasal swabbing missed more that half of the MRSA cases and thus the high NPV was almost exclusively driven by the extremely low probability of MRSA pneumonia in their cohort of cases. Should this test be used in a population with ten times the prevalence, the conclusion would be very different. 
Before moving to a point-by-point proposal for minor revisions, I should also state that in my opinion the title of manuscript is misleading and not really linked to their primary conclusion (which according to their Methods is the falling MRSA prevalence through the study period).
Below you will find some point-by-point comments:
Line 34: consider rephrasing “we frequently face with starting an empiric therapy”
Line 41: There is some mistyping here.
Line 43: please rephrase
Line 54: Replace “was” with “were”
Table 1, 1st column: consider aligning left, because central alignment creates some confusion on whether a particular row is a subcategory of the row above.
Table 2, 1st row , 1st column: please correct Ccreening
Table 2, last column, header: update to Total
Line 95: correct sensibility to sensitivity.
Line 119: please rephrase “only a few numbers of VAP”
Line 152: probably you mean positive results.
Line 220: correct “america” to “America”

Reviewer 2 Report

Authors reported an interesting experience about the role of MRSA or MSSA in VAP in a single center ICU. Authors concluded that having VAP due to MRSA did not have any impact on LOS and mortality. Moreover, the MRSA nasal-swab testing demonstrated a very high negative predictive value for MRSA VAP. Results suggested the potential value of a patient-centered approach to improve antibiotic stewardship. 

In my opinion the study is well-written and methods are appropriate. I have only minor observations:

  • data about mortality and LOS should be avoided. The study is mainly focused on the role of colonization and the NPV of nasal swab. I think that analysis of mortality should be delated or, conversely, deepened. Authors should report clinical data, therapeutic approach, etc.
  • discussion is weak. Authors should discuss previous studies and discuss their data adequately
  •  in methods authors should state if in their institution is used a decolonization program, especially in ICU

Overall, I think that the manuscript could be published is adequately revised.

Round 2

Reviewer 1 Report

Dear Editor,

Regarding my first point I am afraid that I have failed to make my self clear. The Authors are right in their response that BAL is probably a valid strategy in diagnosing VAP. However, not every hospital has adopted bronchoscopy in VAP diagnosis. So, if research is based on BAL in a hospital with low bronchoscopy rates, a significant amount of (by other means diagnosed) VAP cases would not have been included (i.e. ascertainment bias). The references provided by the Authors are indeed useful, but rather not relevant to my comment. The Author’s need to state that this research is performed in a unit where bronchoscopy is indeed their standard-of-care in diagnosing VAP (by the way, a description of the ICU in general would be welcome, e.g., how many beds, how many admissions per year, surgical/medical etc.).

With regards to my second point, the Authors chose the least preferred option, that is rightfully downgrading the statements instead of a proper multivariate testing. Despite the paper looking “emptier” now, it is probably more properly modestly worded.

Third, by reading the Authors’ reply to my third main point, it is now clear to me that their key statement is a critique to the guidelines stating that whenever your MRSA/total S.aureus ratio is greater than 0.2-0.25, each time you diagnose VAP you should prescribe anti-MRSA treatment. The Authors are essentially stating that “Gentlemen, our MRSA/total ratio was over 0.2 for each consecutive year, for 10 years now. However, if we followed your guidance, we would have to treat some 1500 patients with MRSA drugs just to cover the eventual 41 cases”. I agree that this is a correct statement, however since it is only the low absolute prevalence that drives this conclusion, I believe that the type of the Article should be a letter/short communication rather than a formal Article (by the Journal’s definition). I am afraid that the outcome comparisons and the nasal colonization topic do little to increase the weight of the article. If the Journal does not accept letters or short communications, the paper should be more properly suited for a Journal that does so.

Now regarding the nasal colonization, I think that the Authors results point to the opposite than the desired direction.  Indeed, if we were to reject the guidelines guidance by the nasal testing, we would have missed half of the MRSA cases. Thus, I don’t know what “Should guide empiric therapy” (as is the paper title) but by the Authors’ numbers, this should not be nasal colonization (though other research which is mentioned by the Authors point otherwise). If the Authors choose to retain it in the paper, then they must discuss the reasons why nasal colonization failed.  

In this regard, the conclusion “Furthermore, we suggested the potential value of a patient-centered approach in order to improve antibiotic stewardship” is not justified, since the Authors have not identified any patient characteristic that should guide empiric MRSA coverage.

Last, the Authors have not replied on my concerns over a misleading title.

Author Response

With this further reply letter we hope that the still ambiguous items are now clarified, and we would like to express our appreciation for the careful and improving approach we have gained with this peer review.

  1. Regarding my first point I am afraid that I have failed to make my self clear. The Authors are right in their response that BAL is probably a valid strategy in diagnosing VAP. However, not every hospital has adopted bronchoscopy in VAP diagnosis. So, if research is based on BAL in a hospital with low bronchoscopy rates, a significant amount of (by other means diagnosed) VAP cases would not have been included (i.e. ascertainment bias). The references provided by the Authors are indeed useful, but rather not relevant to my comment. The Author’s need to state that this research is performed in a unit where bronchoscopy is indeed their standard-of-care in diagnosing VAP (by the way, a description of the ICU in general would be welcome, e.g., how many beds, how many admissions per year, surgical/medical etc.).

Reply: Thank you. It is now clearer to us what was meant. Accordingly, we have added the details you suggested in the further modified version of the paper. In particular, it is now specified that BAL is the standard-of-care in diagnosing VAP. In addition, we have added a brief description of our intensive care, as you requested.

Specifically, our ICU has 24 beds, with a total of 1000 admissions/year. Among them, surgical patients account for 25-30% of admissions.  

  1. With regards to my second point, the Authors chose the least preferred option, that is rightfully downgrading the statements instead of a proper multivariate testing. Despite the paper looking “emptier” now, it is probably more properly modestly worded.

Reply: As previously stated, we agree with the reviewer on the appropriateness of the proposed revision.

  1. Third, by reading the Authors’ reply to my third main point, it is now clear to me that their key statement is a critique to the guidelines stating that whenever your MRSA/total S.aureusratio is greater than 0.2-0.25, each time you diagnose VAP you should prescribe anti-MRSA treatment. The Authors are essentially stating that “Gentlemen, our MRSA/total ratio was over 0.2 for each consecutive year, for 10 years now. However, if we followed your guidance, we would have to treat some 1500 patients with MRSA drugs just to cover the eventual 41 cases”. I agree that this is a correct statement, however since it is only the low absolute prevalence that drives this conclusion, I believe that the type of the Article should be a letter/short communication rather than a formal Article (by the Journal’s definition). I am afraid that the outcome comparisons and the nasal colonization topic do little to increase the weight of the article. If the Journal does not accept letters or short communications, the paper should be more properly suited for a Journal that does so.

Reply: We have already argued about the content of the reviewer's third point, applying the requested changes in the text of our revised paper. We understand the reviewer's argument, but we feel that the article may be worthy of an original article rather than just a letter. However, such a judgement might be maybe left to the Editor, once he or she has read the work and the revisions applied. If a more restrained version of the paper is desired, we will be pleased to summarise our results in a briefer text, although we believe that the current text is more clearly and concisely expressed.

  1. Now regarding the nasal colonization, I think that the Authors results point to the opposite than the desired direction.  Indeed, if we were to reject the guidelines guidance by the nasal testing, we would have missed half of the MRSA cases. Thus, I don’t know what “Should guide empiric therapy” (as is the paper title) but by the Authors’ numbers, this should not be nasal colonization (though other research which is mentioned by the Authors point otherwise). If the Authors choose to retain it in the paper, then they must discuss the reasons why nasal colonization failed.  In this regard, the conclusion “Furthermore, we suggested the potential value of a patient-centered approach in order to improve antibiotic stewardship” is not justified, since the Authors have not identified any patient characteristic that should guide empiric MRSA coverage.

Reply: We apologize for not having properly edited this issue, which we realize is still misleading to the reviewer. We have therefore amended both the title of the paper and the text.

We wish to emphasize that nasal swabbing for MRSA is part of the infection control policies performed in ICU, but we do not think it should be used to guide empirical therapy. We believe we already demonstrated in the previous reply letter and in our text how we are not claiming an universal implementation of MRSA nasal screening to start an empirical antibiotic therapy against MRSA.

Instead, data on MRSA nasal colonization only complete our findings, and enrich our results. In our opinion, without reporting these data, the paper would have been open to considerable critique, as it would have ruled out one of the major risk factor for MRSA infection.

Nevertheless, as this issue is still ambiguous to the reader, we have amended the headline as advised, and removed the sentence "Furthermore, we suggested the potential value of a patient-centred approach in order to improve antibiotic stewardship" from the body of our revised paper.

  1. Last, the Authors have not replied on my concerns over a misleading title.

Reply: We modified the title.

Round 3

Reviewer 1 Report

Dear Editor,

I still believe that the strength of the paper is the epidemiologic description of a falling MRSA prevalence among VAP cases. Apart from that, the other sections of the manuscript are weak. The between group comparisons are totally unadjusted for confounders while the nasal colonization data are of some significance, but mishandled in the Discussion. 

As a result the paper is of low scientific coherence, with the three parts above been loosely connected and not serving a clearly defined rationale. 

Based on the above I still thing that the paper should be abbreviated and, unfortunately, have to reject it in its current form.
